# Can Vision Models Mirror Human Understanding of Increasing Task Difficulty?

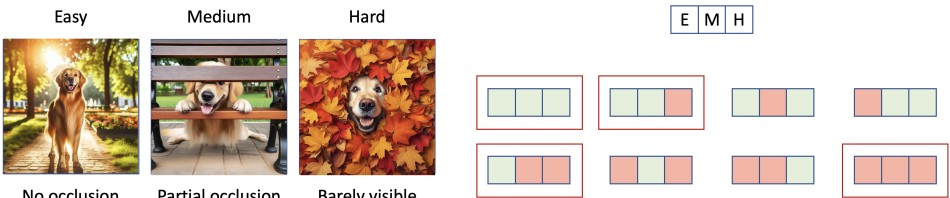

Figure 1: **Left:** Sample images from our proposed test set generated using GPT + DALL-E 3. For the class of *golden retriever* and attribute *occlusion*, we generate images of varying difficulty. Intuitively, it is easier to classify the leftmost image as *golden retriever* compared to rightmost image. **Right:** Possible responses (correct/incorrect) of a model on the easy/medium/hard image on the left side. *Cannot solve easy one → cannot solve difficult one. Can solve difficult one → can solve easy one*: this hypothesis is only satisfied in 4 (in red) out of 8 possibilities.

## Abstract

When a human undertakes a test, their responses likely follow a pattern: if they answered an easy question $(2 \times 3)$ incorrectly, they would likely answer a more difficult one $(2 \times 3 \times 4)$ incorrectly; and if they answered a difficult question correctly, they would likely answer the easy one correctly. Anything else hints at memorization. Do current visual recognition models exhibit a similarly structured learning capacity? In this work, we consider the task of image classification and study if those models' responses follow that pattern. Since real images aren't labeled with difficulty, we first create a dataset of 100 categories, 10 attributes, and 3 difficulty levels using recent generative models: for each category (e.g., dog) and attribute (e.g., occlusion), we generate images of increasing difficulty (e.g., a dog without occlusion, a dog only partly visible). We find that most of the models do in fact behave similarly to the aforementioned pattern around 80-90% of the time. Using this property, we then explore a new way to evaluate those models. Instead of testing the model on every possible test image, we create an adaptive test akin to GRE, in which the model's performance on the current round of images determines the test images in the next round. This allows the model to skip over questions too easy/hard for itself, and helps us get its overall performance in fewer steps.

## 1 Introduction

Imagine a math teacher grading a student's answer sheet, and finds that they got the answer of $(2 \times 3)$ wrong but the answer for $(2 \times 3 \times 4)$ right. The teacher will rightly wonder whether the student properly learnt the concept of multiplication or whether they memorized the answer to the more difficult question. This is because there is a characteristic way in which humans learn any concept: if they cannot answer an easy question, they very likely *cannot* answer a more difficult one. And conversely, if they can answer a difficult question, they most certainly *can* answer an easier one as well. Neural networks are also trained to learn concepts to perform a task. Do they also learn those concepts in a similarly characteristic way?

In this work, we study this question in the context of vision models, specifically for the task of image classification. Our goal is to see if modern visual recognition systems (e.g., ConvNext (Liu

et al., 2022), ViT Dosovitskiy et al. (2020)) have such human-like behavior to easy/hard-to-classify images. We don't think the answer to this question is straightforward because vision models typically are not explicitly trained to perform well on hard *only* if they perform well on easy, unlike humans who learn through a well defined curriculum. If these models do indeed mimic humans, it will only be an emergent property. Since there does not exist appropriate real datasets labeled with ground-truth difficulty, we *generate* one instead. Recent image generative models have become capable of generating very high quality images (Rombach et al., 2022), good enough to be used in training recognition models (Yu et al., 2023; Azizi et al., 2023), and we believe that they are good enough to be used for our evaluation. With the aid of recent large language and generative models (GPT-4 Achiam et al. (2023) + DALL-E 3 Ramesh et al. (2021)), we design a prompting system to generate descriptions of images of three levels of difficulty. For example, *an image of a fully visible dog* and *an image of dog only partly visible* can be considered to be image descriptions of an easy- and hard-to-classify images respectively. We use DALL-E 3 to take in these different difficulty level prompts and generate images while faithfully preserving the desired attributes. Fig. 1 (left) gives an example.

Once we have the easy, medium and hard-to-classify test images, we record if the model predicts the class correctly or incorrectly. Fig. 1 (right) depicts the 8 possibilities of model's behavior (green/red represent correct/incorrect response). If the model truly learns to classify images by developing the aforementioned notion of easy/hard concepts, then its responses should fall under 4 out of the 8 possibilities highlighted in a red box. Our first key finding is that, for most of the current visual recognition models, their responses *do indeed* fall under the 4 highlighted categories around 80-90% of the time. This result hints that even without an explicit supervision, visual recognition models learn to learn things in a structured way.

While an intriguing result in its own, we believe that this can have applications, especially in the way we evaluate models. We take inspiration from how students are often tested using standardized tests, like the Graduate Record Examination (GRE), for admissions into U.S. universities. These tests are adaptive in nature, where questions in the next round depend on how well the student does in the current one. So, for example, if the student cannot solve easy-medium questions, there is not much point giving them difficult questions in the next round; i.e., one can reliably *predict* that they will get zero points for those hard questions. We develop a similar GRE-type test to evaluate visual recognition models on the generated dataset proposed above. The test is broken up into multiple rounds. In the first round, the model is shown images of medium difficulty on average. Its score in this round determines the distribution of easy/medium/hard questions in the next round. That is, similar to GRE, we can skip over images that are too easy/hard for the model to classify. Thus, instead of evaluating the model on every possible image in the test set, this way of dynamically selecting the images helps approximate that total score of the model on the whole set using only 25% of the test images.

Additionally, the newly proposed dataset can have usefulness in and of itself. We generate images from 100 categories taken from ImageNet (Deng et al., 2009). For each category, we consider 10 attributes. Within each attribute, we generate 12 images for 3 levels of difficulty, bringing the total number of images to 36,000. However, different from standard benchmarks like the ImageNet validation set, these 36k images are labeled with attribute value, difficulty, in addition to the ground-truth class. This can enable analysing models on a much finer level (e.g., ResNet-50 struggles to detect dogs from a side view).

In summary, our work has the following contributions. We present a new method to study the learning dynamics of modern visual recognition systems using the concept of example difficulty. To do this, we create a new test set of synthetic images labeled with class, attribute, and difficulty level. Our results indicate that most of the models do in fact develop a semantically meaningful notion of example difficulty while learning visual concepts, without having access to any external supervision. Using this newly found property, we develop a multi-round adaptive test, inspired by GRE, which steers the future test images according to a model's ongoing performance. This facilitates skipping over too easy/hard questions, and helps assess a model's performance using a fraction of test images.

## 2 RELATED WORK

**Neural network learning mechanisms.** Understanding how neural networks learn concepts has long been a central theme in deep learning. Prior work has examined their tendency to generalize versus memorize, even under random labels (Zhang et al., 2021; Arpit et al., 2017), and suggested that generalization depends more on data than model capacity (Dinh et al., 2017; Krueger et al., 2017). Another perspective comes from feature-importance visualization, such as gradient-based methods (Simonyan, 2013) or CAM (Zhou et al., 2016), which highlight regions critical to a model's decision. Training dynamics also reveal biases: networks tend to prioritize easy-to-learn features like texture (Geirhos et al., 2018), while neglecting harder features such as shape (Geirhos et al., 2020) or minority samples (Mehrabi et al., 2021). These insights motivate curricula that present tasks of increasing difficulty (Bengio et al., 2009; Saxena et al., 2019), but such methods neither guarantee that harder concepts are learned only after easier ones nor prevent forgetting of earlier-learned concepts. To our knowledge, we are the first to ask whether this principle instead emerges naturally in neural networks after full training.

**Datasets for studying models' properties.** The standard evaluation of image classifiers is accuracy on human-collected test sets, but benchmarks such as ImageNet (Deng et al., 2009) have raised concerns of saturation (Mayilvahanan et al., 2023). To address this, new datasets test robustness under distribution shifts (Recht et al., 2019; Wang et al., 2019; Barbu et al., 2019; Hendrycks et al., 2021; Taesiri et al., 2024; Hendrycks & Dietterich, 2019; Geirhos et al., 2018), while others employ synthetic data. For example, the Photorealistic Unreal Graphics (PUG) dataset (Bordes et al., 2024) leverages Unreal Engine to probe factors like pose, texture, and lighting; ImageNet-D (Zhang et al., 2024) uses Stable Diffusion to generate challenging images; and Spawrious (Lynch et al., 2023) targets spurious correlations. Parallel efforts study image "difficulty," defined at the class level (Barbu et al., 2019) or per image via human response times (Mayo et al., 2023), model agreement (Meding et al., 2022), or scaling-based accuracy estimates (Jiang et al., 2021). Yet these difficulty notions are isolated and not tied to interpretable attributes. We argue that difficulty is best understood by considering what would make an image easy—for instance, an occluded dog (Fig. 1) would be easy to classify if unobscured. Guided by this view, we propose a generative approach to build datasets annotated with explicit, attribute-based difficulty labels.

**Adaptive model evaluation.** Inspired by computerized adaptive testing (CAT) (Van der Linden & Glas, 2000), we propose adaptive testing algorithms for image classification benchmarks. Like CAT, which estimates ability from a subset of questions, our framework reduces computational cost by evaluating models on carefully selected samples while preserving assessment accuracy. Unlike prior lifelong evaluation frameworks that sub-select existing samples via dynamic programming (Prabhu et al., 2024), our approach generates unseen images with DALL-E, ensuring exposure to genuinely new data, mitigating leakage, and improving the reliability of generalization assessment.

## 3 MODELS' BEHAVIOR ON EASY → HARD IMAGES

Humans learn concepts progressively: solving harder problems is only possible after mastering easier ones (Zacks & Tversky, 2001; Newtson, 1973). We investigate whether image classification models exhibit a similar behavior. In Sec. 3.1, we describe our process for generating test data with images of varying difficulty, analogous to human exam questions. In Sec. 3.2, we introduce a method for analyzing model responses to these images to assess whether they mirror human-like learning progression.

### 3.1 DATASET CREATION

In image classification, the standard dataset format is pairs of images and ground-truth labels, $\mathcal{D} = \{(x_1, y_1), (x_2, y_2), ...\}$. For our purpose, however, it is not enough to know that $y_i$ is the correct label for $x_i$; we also require a measure of how difficult it is to classify $x_i$ as $y_i$. Thus, our first step is to formalize this notion of difficulty in the context of our problem.

#### 3.1.1 UNDERSTANDING SAMPLE DIFFICULTY

Consider Fig. 2, specifically the triplet of images in the top right. Each depicts a golden retriever, yet it is clear to humans that the leftmost image is easiest to classify, while the rightmost is hardest, due

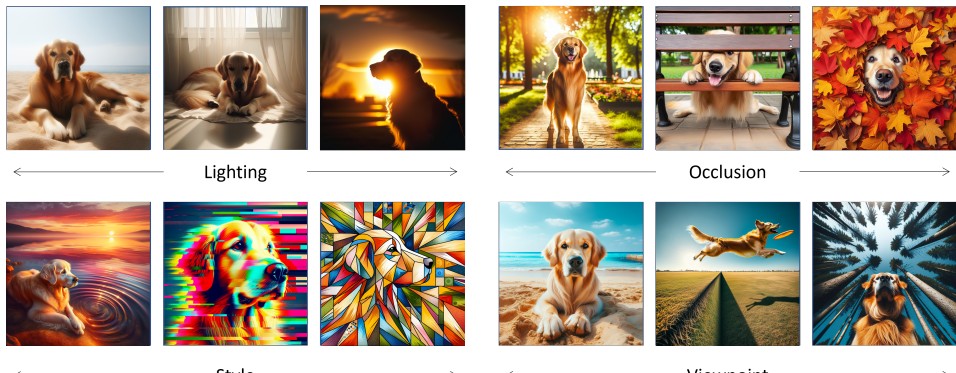

Figure 2: **Visualizing the difficulty of test samples.** All of the images are generated using our proposed pipeline. In each quadrant, we focus on one attribute (e.g., lighting, in the top left), and from left to right we show the images becoming progressively more difficult to be classified correctly.

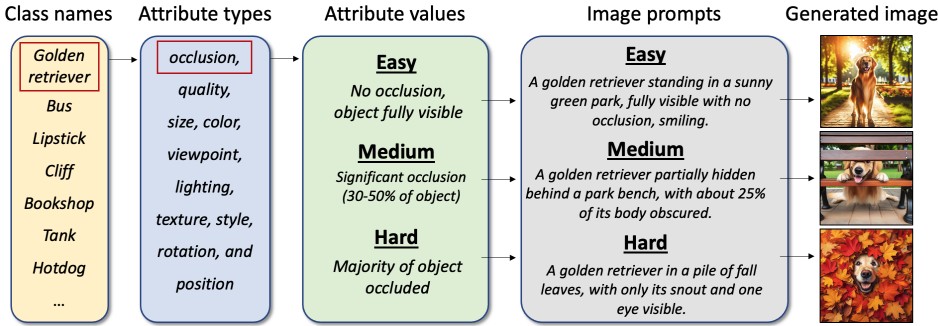

Figure 3: **Overview of the test set generation process.** The first step is to collect the names of the image categories that we wish to test the models on. We then prompt GPT-4 to generate the appropriate attribute values for those categories with various levels of difficulty. Using those, we again prompt GPT-4 to generate text prompts for a category (golden retriever), attribute (heavy occlusion) combination. Finally, we use DALL-E 3 to generate the corresponding images.

to increasing occlusion. Other triplets similarly illustrate an easy → hard progression along different attributes. The key observation is that difficulty is best understood relative to a specific attribute (e.g., occlusion). However, to our knowledge, no real-image dataset provides human annotations of sample difficulty in this manner.

Given recent advances in generative models, we propose to *synthesize* images with specific attributes. Modern text-to-image systems can now produce high-quality images (Rombach et al., 2022; OpenAI, 2024a), to the point of being successfully used for training classifiers (Yu et al., 2023; Azizi et al., 2023). Moreover, text prompts allow precise control over attributes, enabling the generation of images that are easier or harder to classify. We therefore leverage these models to design a controlled evaluation setup.

### 3.1.2 OVERALL IMAGE GENERATION PIPELINE

To generate images of varying difficulty via text, we require three components for each prompt: (i) a class name (e.g., golden retriever), (ii) an attribute type (e.g., occlusion), and (iii) a difficulty level for that attribute (e.g., hard). The combination of (ii) and (iii) specifies an attribute value. For example: "An image of a golden retriever heavily occluded by a door"—here golden retriever is the class, and heavily occluded corresponds to the hard difficulty for the occlusion attribute. Our first step is to collect the set of classes to evaluate. We select 100 object categories from the 1000 ImageNet classes (see Appendix .10 for the complete list). The next step is to define attribute values, such as heavily occluded in the example above, for each attribute–difficulty pair.

**Generating the attributes.** We first prompt GPT-4 (Achiam et al., 2023) to list 10 common attribute types useful for describing image content (see second column of Fig. 3). For each attribute, we then ask GPT-4:

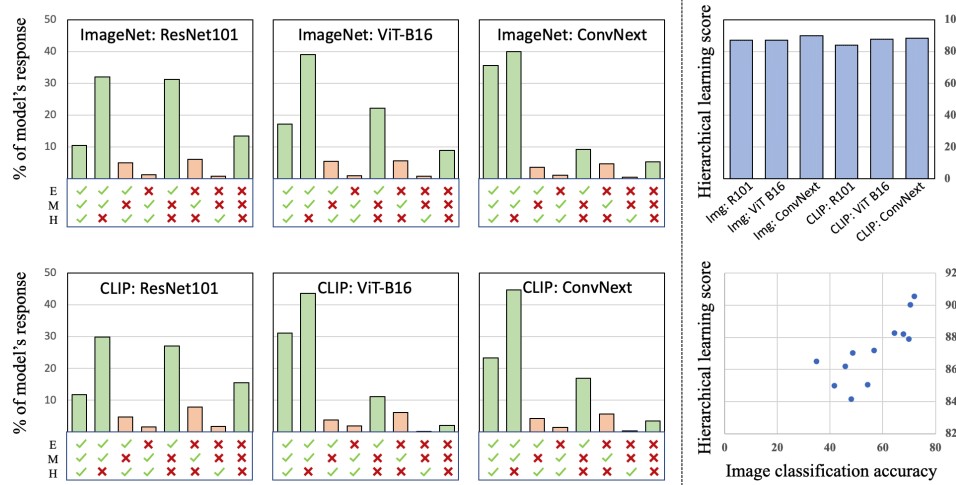

Figure 4: **Top:** Plots depicting % of model's behavior on 12k triplets over the 8 possible patterns of **E**asy, **M**edium, **H**ard. Bars corresponding to principle-following pattern are colored green; others, red. All models behave according to the hierarchical learning principle. **Bottom left:** Hierarchical learning score of 6 vision models. Most achieve a score higher than $85\%$. **Bottom right:** Scatter plot of top-1 accuracy on our test set vs hierarchical learning score of 12 models. PCC value is 0.77.

```
``To generate text prompts for DALL-E that will generate images
of varying difficulty levels for vision models to classify,
please create nine levels of difficulty based on <attribute name>
attributes and group the nine levels of difficulty into categories
of easy, medium, and hard.''
```

GPT-4 returns difficulty-varying attribute values. For example, along the *occlusion* attribute: (i) Easy: No occlusion, object fully visible"; (ii) Medium: Significant occlusion (30–50%)"; (iii) Hard: "Majority of object occluded (70–90%)".

**Generating text prompts.** Using these attribute values, we prompt GPT-4 once more to produce natural descriptions for DALL·E 3. Each description combines a class and an attribute value; e.g., combining *golden retriever* with *heavy occlusion* yields: "A golden retriever in a pile of fall leaves, with only its snout and one eye visible." The overall pipeline is shown in Fig. 3, and Appendix .1 contains detailed prompts.

**Dataset size:** The dataset contains 100 classes, each paired with 10 attributes. For every attribute, we generate 3 difficulty levels with 12 images each, giving: $10 \times 100 \times 3 \times 12 = 36000$ images in total. The dataset is balanced across classes, attributes, and difficulty levels.

### 3.2 HIERARCHICAL LEARNING SCORE

We now describe how to use our easy/medium/hard dataset to test whether models learn concepts hierarchically. Each class combined with its attribute type yields $100 \times 10 = 1000$ pairs (e.g., *golden retriever, occlusion*). For each pair, we construct 12 triplets, each containing one easy, one medium, and one hard image, giving a total of 12,000 triplets.

For each model, we record whether its predictions match the ground-truth class. A model's responses on a triplet fall into one of 8 possible patterns (Fig. 1, right), depending on which of the easy/medium/hard samples it classifies correctly. We then compute the distribution of these 8 patterns across all triplets.

Adopting a human-like hierarchical learning principle—a harder question should be answered correctly only if all easier ones are answered correctly—we note that only 4 of the 8 patterns satisfy this rule (marked in red in Fig. 1). We call these the principle-following patterns. The hierarchical-learning score is defined as the % of a model's responses that fall into these patterns over all 12,000 triplets. The higher this score, the more a model adheres to the hierarchical learning principle.

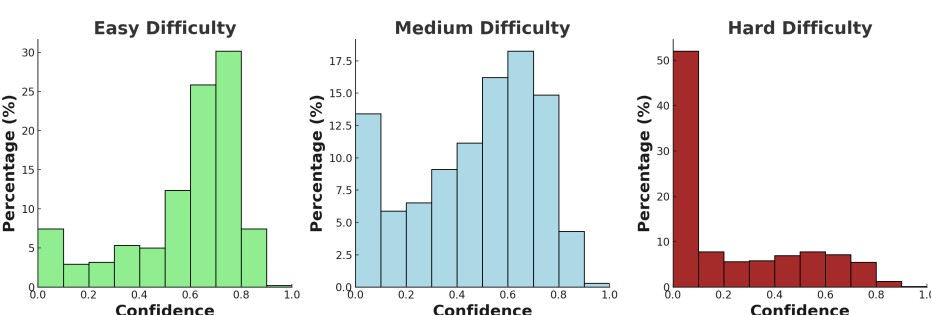

Figure 5: % of our dataset grouped according to classification confidence for the Easy, Medium, and Hard difficulty levels. We average the sample numbers across six selected classifiers (ViT-B16 (Dosovitskiy et al., 2020), ConvNext (Liu et al., 2022), ResNet-101 (He et al., 2016), trained on ImageNet1k (Deng et al., 2009) and LAION (Schuhmann et al., 2022)). See Appendix for more confidence visualization of different classifiers.

**Experiments and Results:** We evaluate three popular architectures—(i) ViT-B16 (Dosovitskiy et al., 2020), (ii) ConvNeXt (Liu et al., 2022), and (iii) ResNet-101 (He et al., 2016)—each trained on ImageNet-1k (Deng et al., 2009) with cross-entropy and on LAION (Schuhmann et al., 2022) with the CLIP objective (Radford et al., 2021), giving six models in total. Results are shown in Fig. 4 (top), where we plot the distribution of behaviors over 12,000 triplets, coloring principle-following patterns in green and others in red. From these distributions, we compute hierarchical-learning scores (Fig. 4, bottom left). Across all models, the majority of behaviors fall under principle-following patterns, yielding hierarchical-learning scores above 85%. In every case, the four most frequent behaviors correspond to the principle-following set, with the most common being Easy: ✓, Medium: ✓, Hard: ×, and the least common Easy: ×, Medium: ×, Hard: ✓. Notably, models trained with cross-entropy on ImageNet and those trained with CLIP on LAION (e.g., ResNet-101 vs. CLIP:ResNet-101) show broadly similar behaviors. These findings indicate that modern visual recognition models follow a human-like hierarchical learning principle—even without explicit supervision to enforce it.

**Model's hierarchical learning score vs accuracy:** Consider two extremes: Model A classifies nearly all triplets as ✓, ✓, ✓, while Model B mostly produces ×, ×, ×. Despite vastly different top-1 accuracies, both would achieve very high hierarchical-learning scores. This raises the question of how the score relates to accuracy. To study this, we evaluate 12 models (6 more than in the previous section; see Appendix) and compute both their hierarchical-learning score and top-1 accuracy. The scatter plot in Fig. 4 (bottom right) shows a clear correlation, with a Pearson coefficient of 0.77. While correlation alone cannot prove that accuracy improves *because* models adopt human-like hierarchical learning, the positive trend suggests that the learning dynamics of strong and weak models are not symmetric: models that perform well in accuracy also tend to score highly on hierarchical learning. Notably, even the lowest hierarchical-learning score observed is 84.2, which we regard as sufficiently high to conclude that all tested models follow the hierarchical-learning principle.

### 3.2.1 EVALUATING CORRECTNESS OF DIFFICULTY LEVELS

To test our hypothesis that vision models follow a hierarchical learning principle, we rely on generated images of varying difficulty (Fig. 2). While these images appear visually plausible, we must ensure that the dataset as a whole meaningfully reflects the intended difficulty levels. For this, we evaluate 36,000 images using six classifiers, analyzing their prediction confidence (softmax probability) at each difficulty tier. We expect high confidence for easy samples, low confidence for hard ones, and intermediate values for medium samples. As shown in Fig. 5, the confidence distributions align with these expectations, validating that our generation process produces images that accurately represent their assigned difficulty levels.

We further conducted a user study to assess how well the generated difficulty levels (Easy, Medium, Hard) align with human perception using pairwise comparisons. From 10 classes with 10 attributes each, we sampled 900 images (3 difficulty levels per attribute). Ten participants evaluated two classes each, completing 540 pairwise comparisons in which they selected the more difficult image; each image appeared multiple times to ensure robustness. Responses were fit with a Bradley–Terry

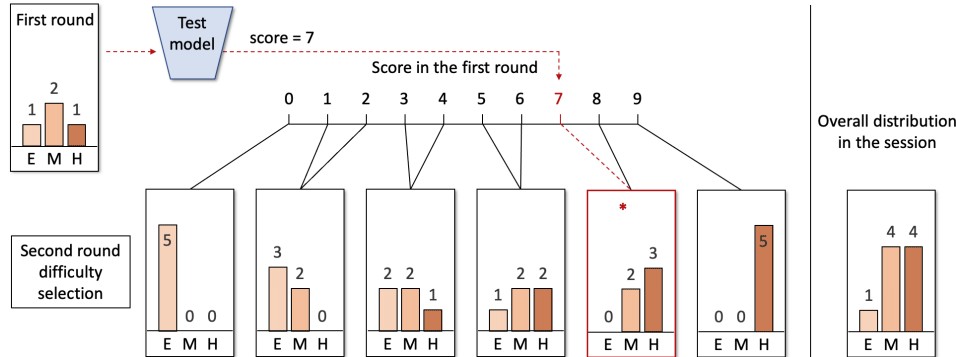

Figure 6: **Adaptive testing of a classifier.** The test involves two rounds. Similar to GRE, the first round is of, on average, medium level difficulty, consisting of 4 test images (1 easy, 2 medium, 1 hard). The model gets a score (max = 9, min = 0) based on which the distribution of images for the next round is chosen. We show an example of a model getting a score of 7 in round 1, because of which in next round there are 0 easy, 2 medium, and 3 hard images. **Right:** Overall, the model gets tested on a total of 9 images; in this case, 1 easy, 4 medium and 4 hard images.

model to derive continuous difficulty scores. The resulting correlations $r = 0.871$ (Pearson), $\rho = 0.883$ (Spearman), and $\tau = 0.749$ (Kendall's Tau), demonstrate strong agreement between human judgments and our assigned difficulty labels (see Appendix Sec. .2 for details).

## 4    ADAPTIVE TESTING OF IMAGE CLASSIFIERS

Humans learn concepts hierarchically, and standardized tests like the GRE exploit this by adapting questions to a student's ability. For instance, if a student struggles with $(2 \times 3)$ or $(4 \times 5)$, it is unnecessary to test them on $(2^2 + 3^2) \times (4 + 5)$—their performance can already be inferred. This adaptive testing avoids asking every easy, medium, and hard question while still accurately assessing ability. In the same spirit, our benchmark of 36k images (spanning 100 categories, 10 attributes, and 3 difficulty levels) enables GRE-style adaptive evaluation of vision models, yielding reliable performance estimates without exhaustively testing all samples.

To evaluate a model, we iterate over all class–attribute combinations (1000 total), each containing 3 difficulty levels with 12 images per level (36 images). The goal is to estimate model performance without testing all 36 images. Inspired by GRE-style adaptive testing, we use two rounds. In round one, the model sees 4 images (1 easy, 2 medium, 1 hard), with scores assigned as 1, 2, and 4 points for correct predictions at easy, medium, and hard levels, respectively. Based on the total score (0–9), round two presents 5 new images with a difficulty distribution determined by the score (Fig. 6). Across both rounds, each model is tested on only 9 images in total, yielding (i) a cumulative score and (ii) accuracies by difficulty tier. While we do not claim this setup is the definitive evaluation scheme, it offers a practical configuration; see Appendix Sec. .3 for further discussion.

After collecting results across all sessions, we can aggregate them in multiple ways. Attribute-level scores are obtained by averaging session scores across 100 classes, while attribute-level accuracies can be computed overall or per difficulty tier. This allows us to identify which attributes a model struggles with (see Sec. 4.3). A global score or accuracy, averaged across attributes, provides an overall performance measure—all derived from only a fraction of the full 36k images.

In the next sections, we discuss how accurate our predictions (score/accuracy) can be when compared against those same values computed over the whole set.

### 4.1    IS THE TEST DIFFERENT FOR DIFFERENT MODELS?

The purpose of adaptive testing is to challenge stronger models with harder samples and weaker models with easier ones. Since each session evaluates a model on only 9 questions, the composition of easy, medium, and hard items will naturally differ across models. Table 1 shows the average distribution of difficulty levels faced by seven models. Most are tested primarily on medium questions,

Table 1: Average number of questions tested per difficulty level for adaptive testing. Models with better performance tend to receive a higher proportion of medium and hard questions, and vice versa.

| Difficulty | Easy | Medium | Hard |
|---|---|---|---|
| ResNet18 | 3.51 | 3.83 | 1.55 |
| ResNet101 | 2.56 | 3.96 | 2.48 |
| ViT-B16 | 2.0 | 3.89 | 3.07 |
| ConvNext-B | 1.55 | 3.46 | 3.98 |
| CLIP-RN101 | 2.60 | 3.89 | 2.51 |
| CLIP-ViT-B16 | 1.47 | 3.61 | 3.93 |
| CLIP-ConvNext-B | 1.62 | 3.79 | 3.58 |

but the balance shifts with capability. For example, ResNet18 (ImageNet top-1 accuracy 69.76%) encounters more easy (3.51) than hard (1.55) samples, while ConvNeXt (top-1 accuracy 84.06%) is tested much more on hard (3.98) than easy (1.55) samples. These results illustrate how models of varying strength trace distinct trajectories of test questions under our adaptive framework.

## 4.2 How closely does Adaptive Evaluation follow Full Evaluation?

We next compare our GRE-style adaptive evaluation to the standard way of evaluating a model on the entire dataset across the ten attributes and three difficulty levels.

To evaluate classification performance, we complement standard accuracy with a GRE-style score that rewards correct answers on harder samples: Score = $(\text{correct}_{\text{easy}} \times 1) + (\text{correct}_{\text{medium}} \times 2) + (\text{correct}_{\text{hard}} \times 4)$, assigning 1, 2, and 4 points for correctly classifying easy, medium, and hard images, respectively (0 for misclassifications). A key advantage of this metric is its ability to distinguish models with similar accuracy. For example, although ConvNeXt-B and CLIP-ViT-B16 achieve close accuracies (70.2 vs. 69.8 in Table 2), their GRE-style scores (59.2 vs. 57.6) reveal which model is better at handling more challenging cases.

Our full generated test set contains 36,000 images—12 images for each combination of 100 classes, 10 attributes, and 3 difficulty levels. We treat evaluation on this complete set, denoted 'Static 12', as the ground-truth baseline. To validate our adaptive testing, we first construct a reduced baseline called 'Static 3', formed by randomly selecting 3 of the 12 images at each difficulty level, yielding 100 (classes) × 10 (attributes) × 3 (difficulty levels) × 3 (images) = 9,000 images in total.

Our adaptive procedure also selects 9 images per class–attribute pair, but distributes them adaptively across difficulty levels rather than fixing 3 per level. This again produces 100 × 10 × 9 = 9,000 images, matching the size of Static 3 but with a different difficulty distribution. We then compare classifier performance on these subsets, aiming for close correlation with Static 12 while reducing error relative to the naive Static 3 strategy.

We compare Static 3 and our adaptive test against the full evaluation (Static 12) in Table 2. Each evaluation is repeated three times, with mean error reported (standard deviations are in Appendix .7). Results show that adaptive testing yields accurate performance estimates using fewer images, and produces smaller errors in both score and accuracy than the Static 3 baseline. This confirms that, leveraging the hierarchical learning behavior of image classifiers, adaptive testing can expedite evaluation without sacrificing reliability.

## 4.3 Detailed error analysis

Because our dataset provides fine-grained labels for attributes such as size, color, lighting, occlusion, and style, we can pinpoint specific failure modes in model performance. Table 3 shows attribute-level results for six classifiers. As observed on the ImageNet validation set (Idrissi et al., 2022), models with similar overall scores tend to align closely on per-attribute scores—for example, ConvNeXt-Base and CLIP ViT-B16 achieve comparable overall performance (59.2 vs. 57.6) and show close agreement on 6 of 10 attributes.

Despite overall improvements, most models consistently struggle with factors such as size, texture, style, and viewpoint, while performing relatively well on object position and image quality. Beyond attributes, our dataset also enables difficulty-level analyses (Tables 5, 6, 7). As expected,

Table 2: Comparing each classifier's score and accuracy using Static 3 and our adaptative method against Static 12 using the Mean Squared Error. Reported errors are averaged over three runs (Std in the supp). Our method provides accurate performance estimates with fewer test images and smaller errors than Static 3, optimizing the evaluation process. (Static #) represents the number of test images used in each level of difficulty for each attribute of a given class.

|  | Classifier | RN101 | ViT-B16 | ConvN-B | C-RN101 | C-ViT-B16 | C-ConvN-B |
|---|---|---|---|---|---|---|---|
| Score | Static 12 | 35.3 | 43.8 | 59.2 | 36.4 | 57.6 | 51.0 |
| Error | Static 3 | 5.4 | 4.6 | 3.2 | 4.5 | 5.5 | 4.8 |
| Error | Ours | 4.2 | 2.7 | 2.5 | 2.2 | 1.5 | 3.3 |
| Acc | Static 12 | 48.5 | 56.9 | 70.2 | 48.1 | 69.8 | 64.6 |
| Error | Static 3 | 4.9 | 2.6 | 2.0 | 4.4 | 3.5 | 1.6 |
| Error | Ours | 3.6 | 1.0 | 0.9 | 3.6 | 2.3 | 0.9 |

Table 3: Score of different models for each attribute. Bold/underline indicates best/second best.

| Attributes | Color | Light | Occlu | Pos | Quality | Rot | Size | Style | Texture | View |
|---|---|---|---|---|---|---|---|---|---|---|
| ResNet101 | 41.1 | 31.1 | 40.2 | 39.1 | 47.0 | 34.8 | 30.4 | 23.7 | 30.4 | 35.2 |
| ViT-B16 | 47.7 | 39.7 | 46.6 | 48.9 | 56.9 | 48.0 | 36.3 | 34.2 | 37.8 | 41.6 |
| ConvNext-B | **63.7** | **60.6** | **59.4** | **86.7** | 66.2 | **77.5** | **40.8** | 41.3 | 42.9 | **52.6** |
| CLIP-RN101 | 39.5 | 33.4 | 33.3 | 62.2 | 36.6 | 39.7 | 34.6 | 30.2 | 23.4 | 31.2 |
| CLIP-ViT-B16 | 62.2 | 58.9 | 46.6 | 85.3 | **67.2** | 59.8 | 39.8 | **53.3** | **50.4** | 52.4 |
| CLIP-ConvNext-B | 53.1 | 49.9 | 48.4 | 79.4 | 57.4 | 47.0 | 36.7 | 45.1 | 41.1 | 47.7 |

performance decreases with difficulty, with attributes like Texture, Style, and Viewpoint showing the steepest drop at the Hard level. An exception is Size: models perform well on easy and medium cases but struggle significantly when many small objects are present.

## 5 DISCUSSION AND LIMITATIONS

The hierarchical learning score captures a different dimension of ability: not just whether a model is accurate, but whether its success reflects a principled learning process, grounded in the way humans learn, rather than something akin to memorization. This learning principle could be studied even for non-vision models, e.g., LLMs/VLMs like OpenAI o1 (OpenAI, 2024b). In fact, the GRE-based adaptive testing will save even more compute cost for such computationally heavy models. Exploring those settings, however, requires redefining "difficulty" in task-specific ways, which is beyond the scope of this paper. Our goal here is to establish the phenomenon clearly in vision, validate it with a controlled synthetic dataset, and provide a proof of concept.

That said, this new score is not immune to extreme cases; a model that fully memorizes test images could still achieve high score. Moreover, while DALL-E 3 enables large-scale synthetic generation with controllable difficulty, it introduces limitations. For rare classes (e.g., African Hunting Dog), DALL-E often produces mislabeled outputs, and for complex prompts (e.g., "A carousel in an amusement park, almost entirely hidden behind a festival tent"), it may generate simplified or unintended scenes, resulting in biased "easy" samples. Although we manually filtered out problematic cases, future work could leverage more advanced generative models to mitigate these issues and improve dataset reliability.

## 6 CONCLUSION

We investigated whether modern visual recognition models display human-like learning behaviors across images of varying difficulty. Using advanced generative models, we created a synthetic dataset annotated with class, attribute, and difficulty labels. Our results show that most models exhibit structured sensitivity to difficulty, even without explicit supervision. Building on this insight, we propose an adaptive testing framework that substantially reduces evaluation time while preserving reliability. Beyond evaluation, the dataset itself—with fine-grained annotations—offers a valuable tool for analyzing model weaknesses. Together, these contributions provide a new perspective on learning dynamics in vision models and introduce an efficient, dynamic approach to model assessment.

**Ethics Statement**   This work does not involve human subjects, personal data, or sensitive information. The synthetic dataset used in our experiments was generated with large language and image generation models (GPT-4 and DALL·E 3), and no copyrighted or private material was incorporated. The dataset contains only artificial images and thus poses minimal privacy, security, or legal concerns. We note that synthetic data can still introduce biases inherited from the generative models; to mitigate this, we constructed a balanced dataset spanning 100 ImageNet categories and explicitly validated difficulty labels both via human judgment and model behavior. The findings of this paper focus on understanding the learning dynamics of vision models and developing more efficient evaluation strategies. We do not foresee direct harmful applications, but we caution that difficulty-aware evaluation frameworks, if misused, could be applied in adversarial ways (e.g., selectively evaluating models to downplay weaknesses). Our contribution is intended to benefit the research community by providing tools for more interpretable and efficient benchmarking.

**Reproducibility Statement**   We have taken several steps to ensure reproducibility of our results. The generation process for the synthetic dataset is fully described in Section 3.1 (Dataset Construction) and Appendix .1, including the prompt design and sampling strategy. All model architectures we evaluate (ConvNeXt, ViT, CLIP, ResNet) are standard publicly available models. The definition of Hierarchical Learning Score (HLS) is given formally in Section 3.2, and the adaptive evaluation protocol is detailed in Section 4. In the supplementary material, we include (i) complete prompts used for dataset generation, (ii) additional experimental results across model families. We will also release the dataset and source code to support full reproducibility.

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

## .1 PROMPT DESIGN FOR IMAGE GENERATION PIPELINE

Please see Fig. 7 for the detailed view of all the prompts used to create the final text caption used by DALLE-3 to generated the images. Note, to achieve finer granularity in difficulty design, we create nine distinct levels, allowing for a more nuanced representation of attribute variation across a spectrum. This finer resolution ensures incremental differences between levels, preventing gaps or uneven difficulty progression. In contrast, directly generating only three broad levels may oversimplify the difficulty range. As shown in the blue box of Fig. 7, we prompt GPT with "group the nine levels of difficulty into categories of easy, medium, and hard" to consolidate them into three final categories.

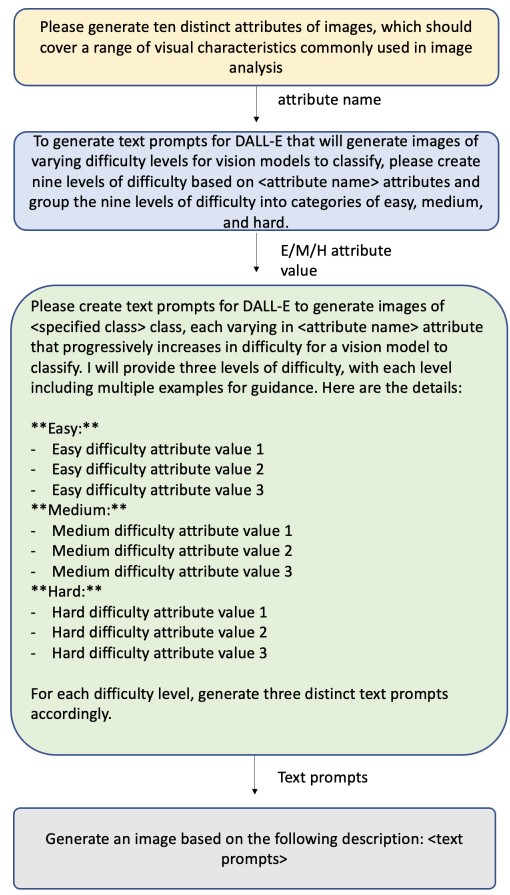

Figure 7: Desinged prompt for test set generation pipeline.

## .2 USER STUDY PIPELINE FOR EVALUATING DIFFICULTY LEVELS OF GENERATED IMAGES

To assess the alignment between difficulty levels of generated images and human perception, we conducted a pairwise comparison study and analyzed the results using statistical correlation metrics. The study pipeline consists of the following steps.

**Dataset and sampling strategy** Our dataset consists of 100 classes, each with 10 attributes and three difficulty levels per attribute (Easy, Medium, Hard), with 12 images per difficulty level. To ensure feasibility for the user study, we randomly sampled a subset of 900 images: 10 classes, 10 attributes per class, 3 difficulty levels per attribute, and 3 images per difficulty level.

**Study design and pairwise comparison setup** To evaluate difficulty perception, participants compared difficulty levels within each attribute rather than across attributes or classes. Each attribute underwent three pairwise comparisons: Easy vs. Medium, Medium vs. Hard, and Easy vs. Hard. Participants were presented with two images at a time and asked: ``Which image is more

difficult to be recognized as [class]?". We show the interface of user study in Fig. 8. Each pairwise comparison consisted of two images, one from each difficulty level. The total number of comparisons was: 27 per attribute, 270 per class, and 2700 across all classes.

**Attribute: occlusion**

Which image is more difficult to be recognized as "brownbear"?

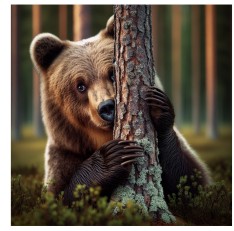 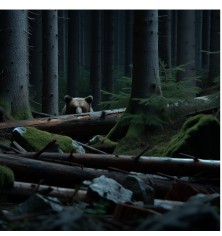

○ Left                     ○ Right

Next

Trial 1 out of 540

Figure 8: **Interface of user study.**

**Experimental setup** We recruited ten participants, each evaluating two classes. Since each class contains 270 comparisons, each participant completed a total of $270 \times 2 = 540$ comparisons. This distribution ensured that all 2700 selected images were evaluated while maintaining overlap across participants to enhance robustness. To reduce bias, image placements (left vs. right) were randomized, and each image appeared in multiple comparisons.

**Analysis and ranking inference** To derive a global difficulty ranking from human responses, we applied the Bradley-Terry Model (BTM), which estimates a continuous latent difficulty score $\lambda_i$ for each image based on pairwise comparisons. Given two images, $i$ and $j$, the probability of selecting $i$ as more difficult is:

$$P(i \succ j) = \frac{e^{\lambda_i}}{e^{\lambda_i} + e^{\lambda_j}} \tag{1}$$

Higher $\lambda_i$ values indicate greater perceived difficulty.

**Correlation analysis: difficulty levels of generated images vs. human-inferred difficulty** To quantify the alignment between difficulty levels of generated images (Easy = 1, Medium = 2, Hard = 3) and human rankings, we computed Pearson correlation ($r$) to measure linear alignment, Spearman rank correlation ($\rho$) to evaluate ordinal agreement, and Kendall's Tau ($\tau$) to assess pairwise consistency. The computed values were:

$$
\begin{aligned}
r &= 0.871 \quad (p < 10^{-280}) \\
\rho &= 0.883 \quad (p < 10^{-296}) \\
\tau &= 0.749 \quad (p < 10^{-187})
\end{aligned} \tag{2}
$$

These results indicate a strong alignment between difficulty of generated labels and human perception.

.3 ABLATION OF HYPERPARAMETERS FOR THE GRE TESTING ROUND

The effectiveness of the adaptive test in approximating overall performance should not be highly sensitive to minor changes in test structure. To validate this, we conducted an ablation study on test parameters by modifying two aspects: (i) the number of questions in the first and second rounds and (ii) the distribution of second-round questions based on first-round performance.

In this modified setting, referred to as *ours new*, the model receives five questions in the first round, consisting of **1 easy, 3 medium, and 1 hard** question, ensuring an average difficulty of medium. The score range is from **0 to 11**. The second-round question distribution, in terms of easy (E), medium (M), and hard (H) questions, is adjusted as follows:

- **Score = 0**    (E=4, M=0, H=0)

Table 4: Impact of different hyperparameters for the GRE testing round. The results of different hyperparameters remain largely consistent.

| Model | Acc (Ours Old) | Acc (Ours New) | Acc (All) |
|---|---|---|---|
| ConvNext-B | 70.5 | 70.8 | 70.2 |
| ViT-B16 | 56.8 | 57.4 | 56.9 |
| ResNet101 | 48.5 | 38.9 | 30.7 |
| CLIP ConvNext-B | 64.8 | 64.1 | 64.6 |
| CLIP ViT-B16 | 70.4 | 69.2 | 69.8 |
| CLIP ResNet101 | 48.0 | 48.8 | 48.1 |

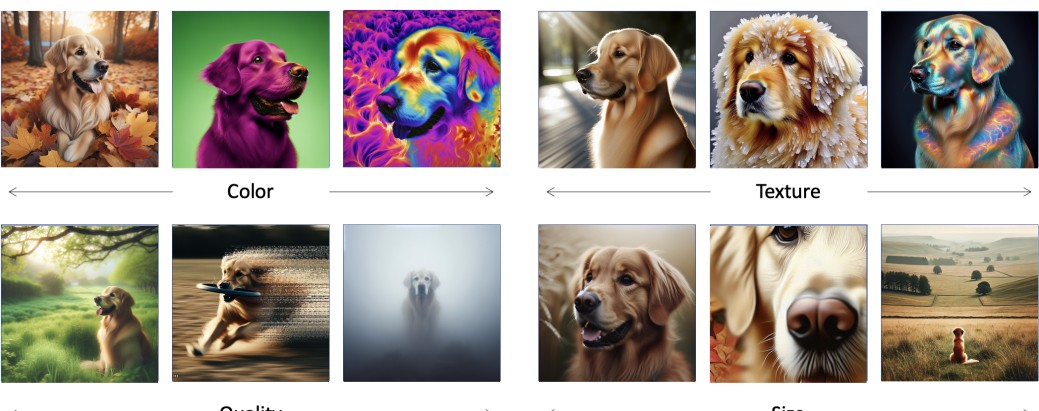

Figure 9: **Visualizing the difficulty of test samples.** All of the images are generated using our proposed pipeline. In each quadrant, we focus on one attribute (e.g., color, in the top left), and from left to right we show the images becoming progressively more difficult to be classified correctly.

- **Score = [1,3]**  (E=3, M=1, H=0)
- **Score = [4,6]**  (E=1, M=2, H=1)
- **Score = [7,10]**  (E=0, M=1, H=3)
- **Score = 11**  (E=0, M=0, H=4)

Using this revised test format, we report model accuracy below, consistent with the results presented in Table 4. For comparison, we replicate the results of the existing adaptive test (*ours old*) alongside the overall accuracy of the static 12-question test. The results indicate that performance remains largely consistent with the previous test version.

### .4 VISUALIZING THE DIFFICULTY OF TEST SAMPLES

We present additional images featuring a golden retriever as the main subject, focusing on attributes such as color, texture, quality, and size. From left to right, the images are arranged to become progressively more challenging for accurate classification. Please see Fig. 9. Finally, we also show more examples for other classes along with their attributes in Fig. 10, 11, 12, 13.

### .5 DETAILED ERROR ANALYSIS

In addition to analyzing attribute-level errors, our generated dataset enables a detailed difficulty-level analysis for each classifier, as shown in Tables 5, Table 6, and Table 7. Across all models, the performance decreases as the difficulty level increases. This is a general trend for each attribute, indicating that all models struggle more with "Hard" samples compared to "Easy" and "Medium" ones. Additionally, attributes like "Texture," "Style," and "Viewpoint" generally have lower accuracies, especially at the "Hard" level. This suggests that these attributes pose more significant challenges for current deep-learning models.

Figure 10: **Visualizing the class of Beer Bottle.**

| Attribute | CLIP ResNet101 | ResNet101 | CLIP ViT B16 | ViT B16 | CLIP ConvNext Base | ConvNext Base | Average (Attributes) |
|---|---|---|---|---|---|---|---|
| Color | 58.89 | 70.74 | 83.33 | 75.56 | **84.07** | 83.70 | 76.38 |
| Lighting | 67.04 | 67.04 | **91.11** | 77.41 | 87.41 | 82.59 | 78.77 |
| Occlusion | 65.93 | 76.67 | 84.81 | 80.00 | **88.52** | 86.67 | 80.77 |
| Position | 97.78 | 96.67 | **100.00** | 97.04 | 99.26 | 97.04 | 97.96 |
| Quality | 69.26 | 78.52 | **89.26** | 80.74 | 87.41 | 87.41 | 82.77 |
| Rotate | 99.26 | 96.67 | **100.00** | 97.78 | **100.00** | 99.26 | 98.49 |
| Size | 98.52 | 97.04 | **100.00** | 98.15 | **100.00** | 99.26 | 98.83 |
| Style | 71.48 | 68.89 | 82.96 | 78.52 | **85.56** | 82.22 | 78.27 |
| Texture | 42.96 | 56.67 | **77.78** | 67.04 | 75.19 | 75.19 | 65.64 |
| Viewpoint | 63.70 | 77.41 | 86.67 | 84.81 | 84.44 | **89.63** | 81.11 |
| **Average** | 73.08 | 77.13 | **89.59** | 83.00 | 89.19 | 88.30 | |

Table 5: Accuracy for different attributes at the easy difficulty level. Bold indicates the highest score, and underline denotes the second highest. The rightmost column shows the average accuracy of each attribute.

## .6 HIERARCHICAL LEARNING SCORE OF ADDITIONAL MODELS

As Section 3.2 mentions Hierarchical Learning Score (HLS), we include an additional six classifiers: ResNet 18, ResNet 50, ConvNext Large, ConvNext Small, ViT Small 16, and ViT Large 16. Their Hierarchical Learning Scores are provided in Table 8.

## .7 STANDARD DEVIATION ACROSS MULTIPLE RUNS

We ran our experiments shown in Table 2 three times. The standard deviation of classification scores, both for ours vs static 3 baseline, is shown in Table 9. We see that the results are consistent (low standard deviation) for all the models.

## .8 MORE CONFIDENCE VISUALIZATION FOR THE EASY, MEDIUM, AND HARD DIFFICULTY

In this section, we visualize the distribution of prediction confidence across the difficulty levels for several classifiers, using our generated dataset. Please see Fig. 14 and 15. We see that they follow a

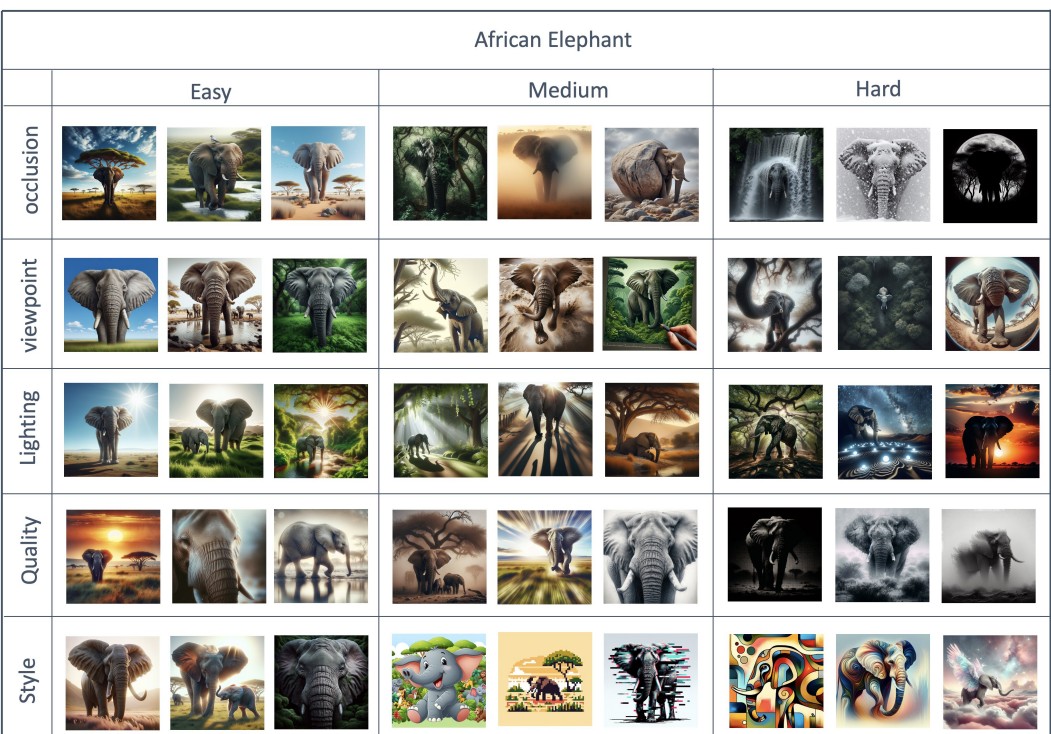

Figure 11: **Visualizing the class of African elephant.**

| Attribute | CLIP ResNet101 | ResNet101 | CLIP ViT B16 | ViT B16 | CLIP ConvNext Base | ConvNext Base | Average (Attributes) |
|---|---|---|---|---|---|---|---|
| Color | 50.37 | 51.48 | 78.89 | 66.29 | 69.63 | **81.85** | 66.42 |
| Lighting | 48.52 | 47.78 | **84.44** | 55.93 | 75.19 | 80.37 | 65.71 |
| Occlusion | 47.41 | 57.78 | 72.59 | 62.96 | 71.48 | **80.00** | 65.37 |
| Position | 67.41 | 38.89 | 93.70 | 54.44 | 91.11 | **94.81** | 73.73 |
| Quality | 43.70 | 60.74 | 78.89 | 67.78 | 75.19 | **77.04** | 67.22 |
| Rotate | 56.67 | 44.44 | 94.07 | 69.63 | 75.19 | **96.30** | 72.05 |
| Size | 62.22 | 54.07 | 81.85 | 70.74 | **85.19** | 85.19 | 73.54 |
| Style | 49.26 | 35.19 | **84.44** | 56.67 | 78.52 | 66.29 | 61.06 |
| Texture | 40.37 | 49.26 | **78.89** | 57.41 | 69.26 | 68.52 | 60.62 |
| Viewpoint | 44.07 | 56.29 | 80.74 | 65.56 | 67.78 | **82.96** | 66.23 |
| **Average** | 50.40 | 49.69 | **82.85** | 62.44 | 75.65 | 81.03 | |

Table 6: Accuracy for different attributes at the medium difficulty level. Bold indicates the highest score, and underline denotes the second highest. The rightmost column shows the average accuracy of each attribute.

similar trend as described in Fig. 5, where the distribution of confidence is progressively decreasing as we move from easy → hard samples.

## .9 ATTRIBUTE OF VARIATION DEFINITIONS

**Position:** The location or placement of the object within the frame of the image. It can indicate whether the object is centered, towards the edge, or even partially out of view.

**Viewpoint:** Describes the angle or perspective from which the object is observed, such as front, side, top-down, or oblique view. The viewpoint affects the amount of detail visible and can reveal or obscure specific features of the object.

**Quality:** Indicates the overall clarity and resolution of the image. High-quality images have fine details and little noise, while low-quality images may appear blurry, pixelated, or noisy, making it harder to discern specific features.

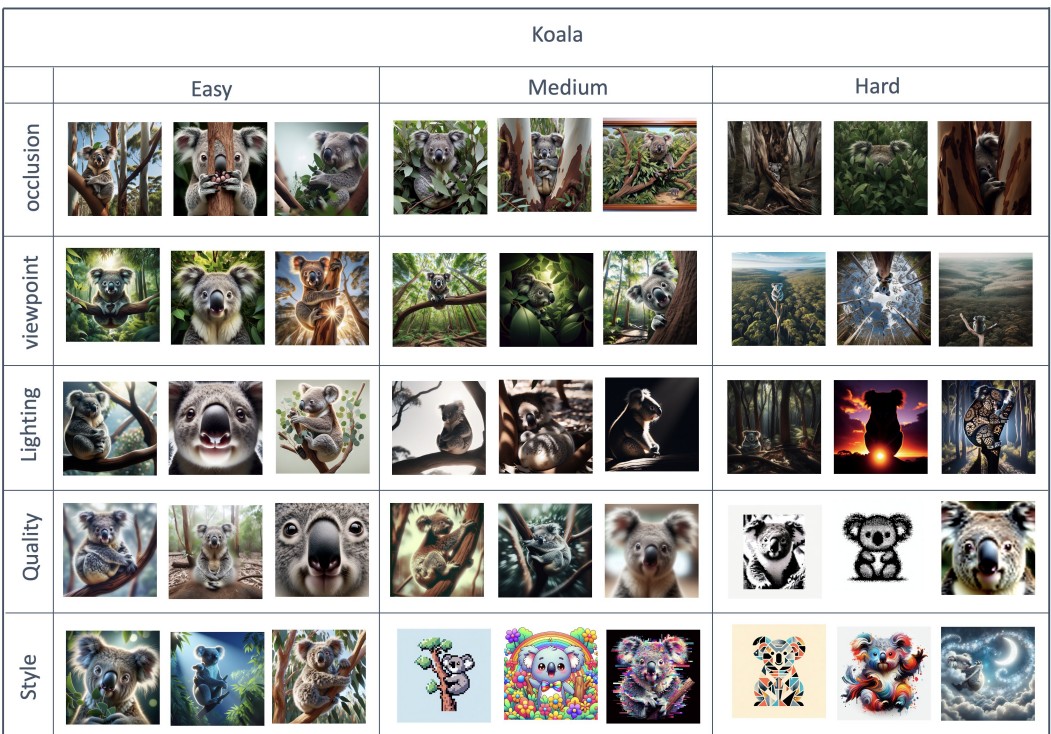

Figure 12: **Visualizing the class of Koala.**

| Attribute | CLIP ResNet101 | ResNet101 | CLIP ViT B16 | ViT B16 | CLIP ConvNext Base | ConvNext Base | Average (Attributes) |
|---|---|---|---|---|---|---|---|
| Color | 29.26 | 28.52 | 48.52 | 31.48 | 37.04 | **47.04** | 36.98 |
| Lighting | 17.41 | 13.70 | 38.15 | 22.22 | 30.74 | **45.19** | 27.57 |
| Occlusion | 18.15 | 22.22 | 24.07 | 30.00 | 26.67 | **44.07** | 27.53 |
| Position | 50.74 | 38.89 | 77.41 | 34.07 | 68.15 | **80.37** | 58.27 |
| Quality | 24.81 | 32.22 | 55.93 | 45.56 | 43.33 | **52.59** | 42.07 |
| Rotate | 16.30 | 14.44 | 32.59 | 24.81 | 19.63 | **62.59** | 28.06 |
| Size | 4.81 | 1.85 | 3.70 | 3.70 | 1.85 | **4.44** | 3.39 |
| Style | 10.37 | 6.67 | 30.37 | 11.85 | 20.37 | **21.48** | 16.52 |
| Texture | 10.00 | 14.44 | 29.26 | 20.74 | 18.52 | **22.22** | 19.20 |
| Viewpoint | 16.67 | 14.07 | **29.63** | 18.89 | 28.51 | 28.15 | 22.26 |
| **Average** | 19.65 | 18.60 | 36.36 | 26.83 | 29.75 | **38.82** | |

Table 7: Accuracy for different attributes at the hard difficulty level. Bold indicates the highest score, and underline denotes the second highest. The rightmost column shows the average accuracy of each attribute.

**Rotate:** Describes the orientation of the object in the image. An object can be upright, tilted, or flipped. The rotation can affect the perception and recognition of the object's standard appearance.

**Occlusion:** Occurs when parts of the main object are blocked or obscured by other objects in the scene. This can make it challenging to identify the full structure of the object.

**Size:** Refers to the object's scale within the image. Size can be influenced by the object's actual size, its distance from the camera, or the zoom level.

**Lighting:** Lighting in the image is either brighter or darker when compared to the prototypical images.

**Color:** Color can indicate the object's natural appearance, the time of day, or the overall mood.

**Texture:** Refers to the surface quality or pattern seen on the object, such as smooth, rough, glossy, or matte.

**Style:** Indicates the visual aesthetics or artistic rendering of the image. This could include photographic styles (e.g., realistic, abstract, cartoonish), drawing styles, or filters applied to the image.

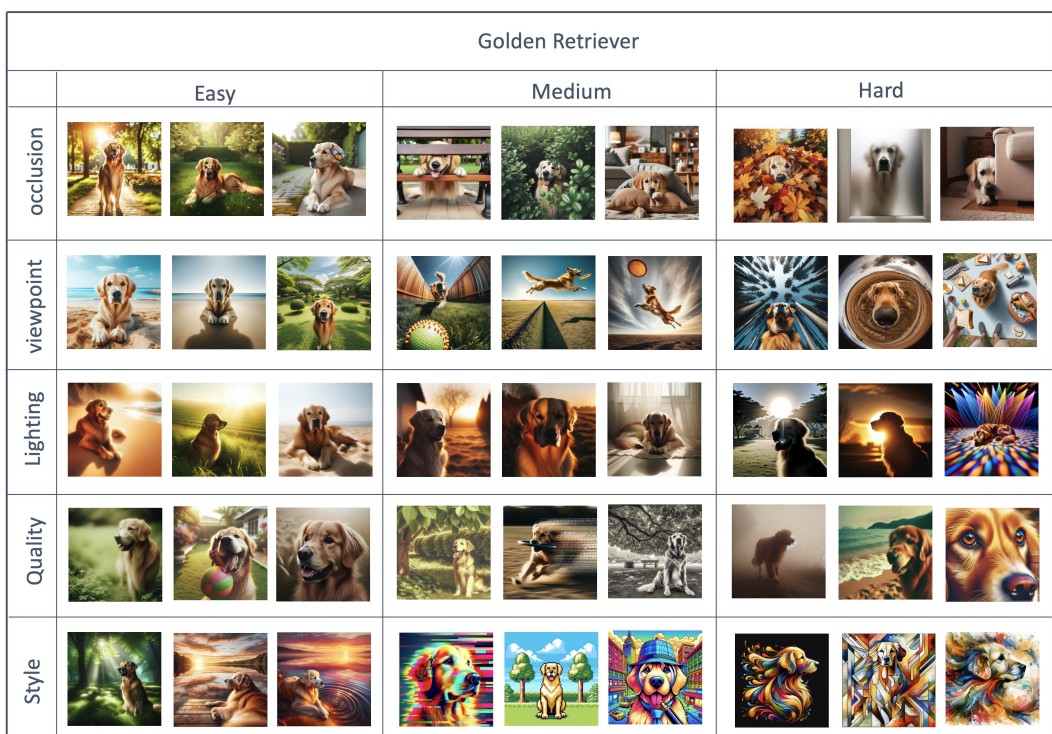

Figure 13: **Visualizing the class of golden retriever.**

Table 8: Hierarchical Learning Score of additional six visual recognition models.

| Classifer | ResNet18 | ResNet50 | ConvNext-L | ConvNext-S | ViT-S16 | ViT-L16 |
|---|---|---|---|---|---|---|
| HLS | 86.52 | 86.19 | 90.56 | 88.22 | 85.00 | 85.04 |

## .10 LIST OF 100 OBJECT CATEGORIES

We selected 100 object categories from the 1,000 classes in ImageNet for our study. These categories represent a diverse range of items, animals, and objects, including: Objects: catamaran, wooden spoon, hourglass, stopwatch, iPod, plate, crate, turnstile, frying pan, comic book, pencil box, cash machine, school bus, obelisk, volleyball, lifeboat, computer keyboard, CD player. Animals: malamute, koala, goose, meerkat, gazelle, bullfrog, loggerhead turtle, box turtle, iguana, Komodo dragon, rock python, diamondback rattlesnake, scorpion, wolf spider, black grouse, flamingo, king penguin, killer whale, Chihuahua, Maltese dog, beagle, Afghan hound, Irish wolfhound, Border collie, Rottweiler, Bernese mountain dog, Dalmatian, Siberian husky, lion, tiger, American black bear, ladybug, fire salamander, hummingbird, goldfinch, toucan, peacock, lobster, Dungeness crab, zebra, bison, hippopotamus, giraffe, kangaroo, platypus, woodpecker, raccoon, skunk, bat, otter, seahorse, jellyfish, sea anemone, coral, stork, crane, tortoise, parrot. Food-related: beer bottle, lipstick, mixing bowl, mashed potato. Others: cliff, black widow, lakeside, sock, great white shark, ostrich, bald eagle, vulture, American alligator, African elephant, golden retriever. This wide range of categories ensures a comprehensive evaluation of model performance across various domains.

## .11 LLM USAGE STATEMENT

We used large language models (LLMs) such as GPT as a general-purpose writing assistant. Their role was limited to aiding clarity, polishing wording, and improving readability of the manuscript. All conceptual contributions, research design, experiments, analyses, and conclusions were developed solely by the authors. The use of LLMs did not influence the research methodology or results.

Table 9: The standard deviation of classification scores for ours vs static 3 baseline.

| Classifier | ConvNeXt Base | ViT B16 | ResNet101 | CLIP ConvNeXt Base | CLIP ViT B16 | CLIP ResNet101 |
|---|---|---|---|---|---|---|
| Std (ours) | 1.3 | 1.9 | 1.1 | 0.7 | 1.0 | 0.8 |
| Std (Static3) | 1.9 | 2.3 | 2.1 | 1.4 | 1.6 | 1.5 |

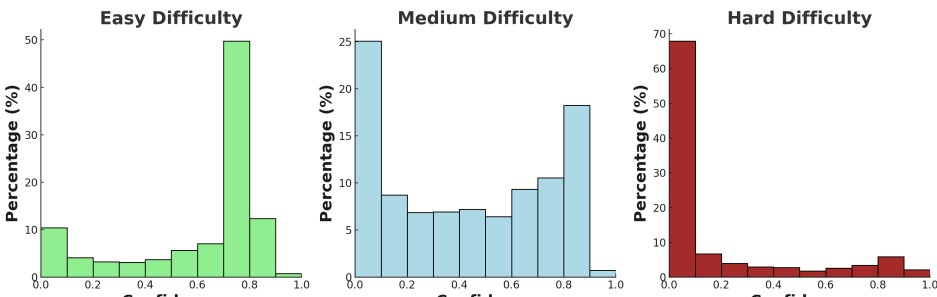

Figure 14: Classification confidence for ViT-B16 model.

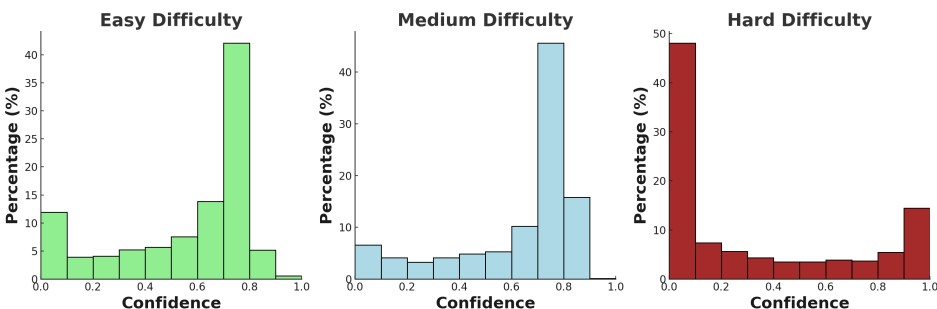

Figure 15: Classification confidence for CLIP-ViT-B16 model.

