# OpenReview forum: "Can Vision Models Mirror Human Understanding of Increasing Task Difficulty?"
_ICLR.cc/2026/Conference — Submitted to ICLR 2026_

### Official Review · Reviewer_tY7R · 2025-10-29

**Soundness:** 2
**Presentation:** 3
**Contribution:** 2
**Rating:** 4
**Confidence:** 3

**Summary:**

The paper studies whether vision models show human-like hierarchical learning. A synthetic dataset of 36,000 images is generated using GPT-4 and DALL·E 3 across 100 classes, 10 attributes, and three difficulty levels. A Hierarchical Learning Score measures whether model predictions follow the easy-to-hard consistency pattern, with most models achieving around 80-90%. Based on this, the authors propose a GRE-style adaptive evaluation that reduces test size while maintaining accuracy, suggesting that vision models display structured, human-like learning behavior and can be evaluated more efficiently.

**Strengths:**

1. The paper introduces a clear and original question on whether vision models exhibit human-like hierarchical learning behavior.

2. The use of GPT-4 and DALL·E 3 to generate an attribute-based dataset with graded difficulty is creative.

3. The adaptive GRE-style evaluation is practical and reduces testing cost.

**Weaknesses:**

1. How is image difficulty defined? In Figure 1, although the leftmost dog has no occlusion and a simple background, its head is much smaller than in the other two examples. Factors such as object size and position in the image may also affect difficulty. How are these aspects systematically controlled or quantified?

2. The concept of difficulty is heuristic and attribute-based, for example, occlusion equals harder and visibility equals easier, but there is no psychometric model linking these levels to human perceptual effort or recognition time.

3. The GRE-inspired adaptive testing framework adjusts subsequent test images based on model performance and relies on predefined difficulty levels. When applying this framework to other datasets, how would these difficulty levels be defined or annotated, and how can the consistency of such labeling be ensured across different data domains?

4. Although the generation pipeline aims to control difficulty for attributes such as lighting, occlusion, style, and viewpoint, other factors like object size, illumination, and background still vary across samples. How can the authors ensure these uncontrolled factors do not influence difficulty, given that such triplets may be confounded?

5. All findings are based on DALL-E 3 images, where potential artifacts, compositional biases, and prompt drift might affect results. The domain gap between AI-generated and real images is not considered and should be tested on real datasets such as ImageNet-C.

6. Human raters evaluated only a subset of the dataset and performed pairwise selections instead of full rankings. What is the reason for this design choice?

7. Human accuracy on the same test set under comparable conditions is missing, which prevents a direct comparison with model performance.

8. The adaptive round thresholds and the 1-2-4 weighting scheme appear arbitrary and lack theoretical justification.

9. Since curriculum learning explicitly enforces an easy-to-hard order, comparing with such training regimes would help place the claim in context.

**Questions:**

1. How is task difficulty defined and annotated? Factors such as object size, position, illumination, and background may all influence difficulty, and it is unclear how these are systematically controlled.

2. The definition of difficulty is heuristic and attribute-based without a psychometric link to human perception. Have the authors compared model behavior with human performance on the same easy/medium/hard triplets to verify whether similar error patterns occur?

3. Although the generation process aims to isolate attributes such as occlusion, lighting, or style, other uncontrolled factors may vary simultaneously. How can the authors ensure that difficulty is not confounded by these changes, given that all results are based on synthetic images with potential artifacts and domain gaps relative to real data?

4. The experimental setup includes partial human validation, arbitrary adaptive weighting (1–2–4), and lacks comparison with curriculum learning or human accuracy on the same test set.

---

### Official Review · Reviewer_JEfw · 2025-10-31

**Soundness:** 3
**Presentation:** 4
**Contribution:** 2
**Rating:** 4
**Confidence:** 4

**Summary:**

The paper describes a new image classification dataset where in addition to the class label, each image comes with a difficulty level classification for several different attributes (such as occlusion, size etc.). Armed with this dataset, the authors explore whether off-the-shelf vision models exhibit performance patterns similar to humans in the following sense: a model is highly unlikely to classify correctly a difficult image if it is wrong on a similar easy image. After confirming that this is indeed the case, the authors develop a dynamic model performance test in the spirit of GRE testing, where the model is tested on difficult images only if it is able to solve similar easy images. This leads to a significant reduction in the number of images a model needs to be evaluated on to accurately estimate its performance on the entire benchmark.

**Strengths:**

The paper is well-written and well-motivated. The methodology for developing the dataset is reasonable. The problem that is addressed is important and the findings are interesting even though they are not surprising to me.

**Weaknesses:**

The main weakness is in the interpretation of the results. The authors suggest that a human-like behavior is emergent capability because models are not trained with an explicit curriculum in mind like humans. A different interpretation is that the patterns can be explained by the fact that models are mainly trained on easy images [1]. They are therefore much more likely to classify an easy image correctly than a difficult one which explains why they are much more likely to follow a human-like pattern. This can also explain the low confidence on difficult images which is simply because they are out of the training distribution.

[1] "How hard are computer vision datasets? Calibrating dataset difficulty to viewing time". Mayo et. al. NeurIPS '23

**Questions:**

Sec 3.2.1 - if the models are not calibrated then it’s not an accurate measure.

---

### Official Review · Reviewer_qhia · 2025-11-01

**Soundness:** 2
**Presentation:** 3
**Contribution:** 1
**Rating:** 2
**Confidence:** 5

**Summary:**

This paper investigates whether vision models exhibit human-like hierarchical learning patterns when classifying images of varying difficulty. The authors generate a synthetic dataset of 36,000 images using GPT-4 and DALL-E 3, spanning 100 categories, 10 attributes, and 3 difficulty levels (easy/medium/hard). They introduce a hierarchical learning score to quantify the pattern of when models can or can't solve hard vs easy problems. The authors find that models achieve 80-90% hierarchical learning scores.

**Strengths:**

- The authors investigate an interesting question, and they present their dataset and experiments clearly
- The dataset and human evaluation could potentially be useful for curriculum learning research

**Weaknesses:**

- The greatest weakness of this work is relying entirely on a generative model to generate images of varying difficulty. How do we know that the types of changes the authors chose actually produce images with the intended difficulty level and attributes?
- This dataset seems to test model robustness to specific kinds of changes rather than difficulty. Similarity datasets have been built in prior works, such as ImageNet-C and ObjectNet.

**Questions:**

- What percentage of generated images fail to exhibit the intended attributes?
- Do your findings transfer to real images with natural difficulty variations that are annotated, such as ImageNet-X or viewing time difficulty (Mayo 2023)?
- Can you explain your process for how you chose your specific prompts used to generate image difficulty variations?

---

### Official Review · Reviewer_FAWK · 2025-11-04

**Soundness:** 2
**Presentation:** 3
**Contribution:** 2
**Rating:** 2
**Confidence:** 4

**Summary:**

This papers aims to study if pretrained visual recognition models for image classification show a similar structured learning behavior with humans (i.e., correctly classifying images follows an order of difficulty from easy to hard). The main hypothesis of the papers is that if a model fails to correctly predict the class of an image, it should also fail on a harder one and vice versa, if it succeeds on a difficult one, it should also succeed on an easier one.

In order to study this, the authors collected a synthetic dataset with generated images using GPT4 and DALLE 3. The dataset contains 100 object classes, 10 difficulty attibutes (e.g., occlusion, size, texture, viewpoint, rotation, etc..) and 3 difficulty levels (easy, medium hard). Each of this was used to prompt GPT4 to produce a text description which was then used to generate the image. The final dataset contains 36k images. The paper shows the results of different image classification architectures (ResNet, ConvNext, ViT) and analyzes the behavior of the models across the difficulty levels. The authors show that their expected hierarchical pattern occurs 80-90% of the time and as they claim this suggests that visual recognition models implicitly learn semantic concepts in a structed way of the task difficulty.

Also, the authors propose an adaptive testing pipeline similar to the GRE exam tests, where the difficulty of test images in one round depends on the model performance in the previous round. Given the same number of images, the paper shows that this adaptive testing estimates better the true accuracy  comparing to a random selection.

**Strengths:**

[S1] The paper poses an interesting question about whether image classification models exhibit human-like learning behavior for difficult levels.

[S2] In general the paper is clearly written and most of the parts are easy to follow. There are clear examples (with multiplications) that can help the reader understand the motivation of this study.

[S3] The authors find empirically that image classification models follow an easy-to-hard pattern 80-90% of the time.

[S4] The idea of an adaptive evaluation setup like the GRE tests is interesting and innovative and can potentially be a new tool for benchmarking models under a limited computation budget.

**Weaknesses:**

The direction of the paper is interesting, but in its current form, there are several major weaknesses that affect the justifications of the paper claims and the overall contributions.

[W1] Definition of task difficulty.
The definition of the task difficulty in this paper is not well-motivated and formulated. It is mainly based on prompting an LLM to list 10 attribute types that describe the image content (e.g., quality, lightining, occlusion, texture, style, etc). Then, the same LLM was prompt to generate text descriptions with varying difficulty (easy, medium, hard) for this selected attribute. This process does not lead to scientifically sound definition for the image classification difficulty and it seems that several important aspects of this difficulty are overlooked (e.g., inter-class similarity and intra-class variation). In the literature, difficulty is either defined as model performance difficulty (how poorly a model performs on a given sample) or human difficulty (how hard an image is for a human to classify). The related work section (L123-135) discuss these alternatives only at a superficial level.

[W2] Generated images.
I understand that using a synthetic dataset with generated images is easier and annotation-free. However, the way that the task is defined and the images are generated do not seem to be able to capture the distribution of real datasets. The studied visual recognition models are pretrained solely on real images. Do you expect a distribution shift with the generated images? When looking at Fig. 10 and 11, it is concerning that the Hard examples are not only hard for the attibute but they looked artificial and out-of-distribution (e.g., pixel art, cartoons, abstract painting-like images). On the other hand, the easy examples look visually more realistic and closer to the ImageNet training data distibutions. L061 states that the authors believe that the generated images are "good enough to be used for evaluation" but the paper does not provide any empirical evidence for that. Therefore, I am not convinced that the the results of the paper reflect only the task difficulty (as it is claimed) and not this distibution shift that occurs from the GPT prompting and the generation. It would be much more convincing if the study was done (fully or partially) on real images.

[W3] Real datasets with difficulty.
The paper claims clearly that no real datasets exist labeled with ground-truth difficulty and that's why the authors generated a synthetic one (L058-059). However, there are several papers in the literature that study the task difficulty from different pespectives for several tasks such as image classification [a,b,c], object detection [b], image segmentation [d] and text classification [e] and provide datasets with real images. It is not clear why the dataset from [a] was not used in this study. This should also be included in the related work section, where only [a] is mentioned.

[a] Mayo et al, How hard are computer vision datasets? Calibrating dataset difficulty to viewing time, NeurIPS D&B 2023
[b] Hendrycks et al, Natural Adversarial Examples CVPR 2021
[b] Ionescu et al, How hard can it be? estimating the difficulty of visual search in an image, CVPR 2016
[d] Vijayanarasimhan et al, What’s it going to cost you?: Predicting effort vs. informativeness for multi-label image annotations, CVPR 2009
[e] Varshney et al, ILDAE: Instance-level difficulty analysis of evaluation data, ACL 2022

[W4] Low classification results.
The classification results on the synthetic images look very low (60-80%) compared to what is expected on real images (e.g., ImageNet) using the pretrained models on only 100 classes. Once again, it is not clear if this is due to the syn-to-real domain shift or the intended difficulty. There is no discussion about this in the paper.

[W5] Inconsistency with numbers.
L 390-391 states that the accuracy of ResNet18 is 69% and the one of ConvNext is 84%. Figure 4 bottom right should show the ConveNext model but the classification accuracy of all models is below 80. Moreover, Table 2 shows that the accuracy of ConvNext is 70.2!

[W6] Closed models.
The synthetic dataset was generated using proprietary models (GPT-4 and DALLE-3) which affects the reproducibility of the dataset. Even if one has access to the prompt design and sampling process, we cannot guarantee the same seed, model version or other hidden changes. Could other open source models be used as alternative (e.g., Stable diffusion or FLUX.1 for image generation and LLaMa, Mistral or Qwen for the text descriptions).

[W7] GRE testing.
Even though the idea is interesting, I do not think that the results of the paper suggest that this is a valid alternative. The MSE in Table 2 and the accuracies in Table 4 suggest that there is up to 17.8% difference in top1 accuracy between the real value and the estimated one. Maybe, this is a mistake in Table 4 for ResNet101, but even for the other models, the estimation is off by up to 0.7%.

[W8] Adaptive testing of a classifier.
It seems that the second round selection distribution is completely arbitrary. Table 4 suggests that the adaptive testing is sensitive to the different hyperparameters. I consider the accuracy difference of more than 1% between Old and New quite high, and therefore leading to an unreliable way to evaluate models.

**Questions:**

For more details please refer to the the weaknesses section above. The most crucial questions regarding the paper refer to the use of generated images and the definition of the task difficulty.

[Q1] Do the generated images capture the distribution of the real images from ImageNet or LAION?

[Q2] Do the Hard vs Easy images capture only the attribute difficulty or the generative models introduce more bias (e.g., pixel art images)?

[Q3] Why are the generated images "good enough for evaluation" (L061)?

[Q3] What is your opinion about the statement in L058-059?

[Q4] Do we want to let GPT-4 define the task difficulty of image classification by proposing 10 attribute categories?

[Q5] How can we consider the GRE testing pipeline as a valid experimental tool that can be used reliably?

---

### Meta-Review · Area_Chair_bdMe · 2026-01-11

**Summary:**

The topic of this paper is interesting, but the meta reviewer agreed with the concerns of the reviewers. Although the authors didn't withdraw the paper, they didn't put any response to the reviewers' comments.

**Reviewer Concerns:**

There is no rebuttal

**Reviewer Scores:**

N/A

---

### Decision · Program_Chairs · 2026-01-26

Reject